# Improvements of Warning Signs for Black Ice Based on Driving Simulator Experiments

**DOI:** 10.3390/ijerph19127549

**Published:** 2022-06-20

**Authors:** Ghangshin Lee, Sooncheon Hwang, Dongmin Lee

**Affiliations:** 1Department of Smart Cities in Graduate School, University of Seoul, Seoul 02504, Korea; joshualee9382@gmail.com (G.L.); tseven37@naver.com (S.H.); 2Department of Transportation Engineering & Smart Cities, University of Seoul, Seoul 02504, Korea

**Keywords:** hazard perception, speeding, road signs visibility, visual behavior, driving behavior, eye tracker, static warning sign

## Abstract

Black ice is one of the main causes of traffic accidents in winter, and warning signs for black ice are generally ineffective because of the lack of credible information. To overcome this limitation, new warning signs for black ice were developed using materials that change color in response to different temperatures. The performance and effects of the new signs were investigated by conducting driver behavior analysis. To this end, driving simulator experiments were conducted with 37 participants for two different rural highway sections, i.e., a curve and a tangent. The analysis results of the driving behavior and visual behavior experiments showed that the conventional signs had insufficient performance in terms of inducing changes in driving behavior for safety. Meanwhile, the new signs actuated by weather conditions offered a statistically significant performance improvement. Typically, driver showed two times higher speed deceleration when they fixed eyes on the new weather-actuated warning sign (12.80 km/h) compared to the conventional old warning sign (6.84 km/h) in the curve segment. Accordingly, this study concluded that the new weather-actuated warning signs for black ice are more effective than the conventional ones for accident reduction during winters.

## 1. Introduction

### 1.1. Background

During winter, black ice is formed on many roadways, and this natural phenomenon introduces many risks. Black ice is a transparent coat of ice on roadways or other transportation surfaces, and it cannot be seen easily. The occurrence time and location of this natural phenomenon are extremely difficult to predict because rapid transitions between dry, wet, snow, and ice are common on such surfaces. This recurrent condition caused by water, ice, and snow causes many accidents worldwide [1,2,3,4,5]. Given the terrain in Korea, many roadways pass through mountainous areas [6]. For numerous reasons, the rate of casualties in black ice-related accidents is 66.2% higher than that in general traffic accidents in Korea [7]. This can be ascribed to the fact that it is difficult for drivers to recognize road freezing. Moreover, the coefficient of friction under such conditions is lower than that under dry and wet surface conditions, which affects the stability of the vehicles and the probability of traffic accidents. The limitation of static conventional black ice warning signs in terms of contributing to drivers’ perception of hazards must be addressed. To evaluate whether the existing warning signs have positive effects on traffic safety, it is essential to know whether they serve their designed purposes. Of all the different components of driving skills, hazard perception alone has been found to be related to accident involvement in several studies [8,9,10,11]. Hazard perception in hazardous driving environments ahead has been studied across different age groups [12,13,14]. In these studies, it was assumed that the conventional static warning signs for black ice do not contribute to hazard perception among drivers of all age and gender groups. Therefore, more effective signs to warn drivers about black ice are needed.

### 1.2. Objective

The primary objective of this study was to investigate the effectiveness of the conventional static warning signs for black ice and develop new warning signs that are actuated by weather conditions. To these ends, driving simulator experiments were conducted using an eye-tracking device. As further highlighted in Section 3.2, the new weather-actuated warning sign for black ice and its final design selected were replicated to the greatest extent possible in terms of realism and detail in the driving simulator, thus allowing participants to provide natural behavioral datasets. More detailed insights into the methodological design of this study are provided later.

## 2. Literature Review

### 2.1. Impact of Warning Signs on Safety

To assess the impact of warning signs on safety, we reviewed related studies on warning signs for black ice. We determined the variables which studies used to evaluate the effectiveness of warning signs for speed reduction. Next, we have reviewed studies of visual behaviors for warning sign detection. Hazard perception in hazardous environments holds great importance for safe driving [9,15]. An understanding of hazards during the early development of driving prevents drivers from being involved in collisions [8,16], However, the conventional static warning signs for occasional hazards may be ineffective because, most of the time, drivers do not perceive any real threat from such signs. In the literature, many operating traffic devices have been evaluated for their effectiveness in terms of hazard warning, but only marginal attention has been dedicated to the effectiveness of conventional static warning signs for black ice. Several studies based on driving simulator experiments have suggested that drivers may become complacent about the importance or meaning of conventional static warning signs [17,18,19,20]. Despite the broad use of conventional static warning signs in the highway system, the effectiveness of these signs in improving drivers’ hazard perception has hardly been investigated. Most studies that investigated the effectiveness of warning signs considered nonconventional warning devices, such as curve warning signs [21,22], flashing LED stop signs and optical speed bars [23], rumble strips [24,25], retro-reflective fluorescent signs, and various road markings as indications of speed limits [20,26,27]. Most researchers examined the effectiveness of nonconventional warning devices relative to that of conventional warning devices by conducting travel speed, acceleration, deceleration, and gas/brake pedal power analysis. Accordingly, more research on conventional static warning signs is necessary.

Generally, many studies on the performance of warning signs have been conducted to determine the effect of these signs on reducing vehicle speed and enhancing road safety in different conditions [28,29,30,31,32]. In terms of curve advisory speeds, many studies have examined the effectiveness of curve warning signs [17,33,34,35]. In terms of the variation of curve warning signs, regardless of their immediate purpose, the literature review showed that warning signs reduce operating speeds in hazardous roadway sections and, thus, affect safety. Moreover, it suggested that typical warning signs are often overlooked owing to their frequent use and, consequently, drivers experience hazard perception latency [36].

For the impact of signs in winter weather conditions, the literature review revealed limited studies. However, all related published reports provided importance of further investigation for the winter weather conditions to prevent black ice-correlated fatalities and injuries [37,38,39,40]. One study found that flashing signals with a reduced speed limit once the sensor system detects ice formation are more effective than static warning signs. The yellow lights start to flash to inform drivers of black ice ahead, and red lights can be used by authorized officers to close lanes or road sections in case of emergency [39]. This study addressed the high cost of black ice-detecting devices as a limitation for application. As mentioned previously, this study analyzed drivers’ speed data obtained from the simulator experiments to determine the effectiveness of static warning signs in terms of reducing vehicle speed and enhancing road safety. After a review of warning signs and driving safety, the travel speed and gas pedal power for acceleration before signs were determined as a measurement of the effectiveness of conventional static warning signs on drivers’ behaviors.

### 2.2. Warning Signs and Visual Behavior

When drivers read signs, they move their eyes from the road to the signs through an alternating pattern of fixations (points at which the eyes are stationary and focused on an object) and saccades (points at which the eyes are moving between fixations). The basic premise of visual behavior is that increased processing demands are related with increased processing time or a change in pattern of fixations [41]. Increased processing time may be indicated by longer duration fixations or a larger number of fixations.

Previous studies have revealed many details about the differences in visual behavior between novice and expert drivers [16,42] and in different visual clutter conditions [43,44]. Even so, considering the limitations of static warning signs for black ice, the research attempted to evaluate how effective the signs were in providing warning information regarding hazards for approaching drivers through measuring visual behavior. In the literature, many studies have attempted to set the hazard perception parameter in terms of standard response time and eye movement analyses. One study calculated braking responses, number of fixations (N), and fixation duration (milliseconds) among experienced and inexperienced drivers. The results indicated that hazard perception was faster among experienced drivers than among inexperienced drivers [45].

In terms of evaluation, many related studies used the number of fixations (N) and fixation duration (milliseconds) as a measurement of the effect of warning signs, along with speed and gas pedal power during acceleration as driver performance variables. According to the literature review, fixation number is a measurement of visual behavior that is strongly correlated with reaction time and, therefore, used as a measure of search efficiency [46]. Average fixation duration is a measure of the amount of time necessary for information processing at each fixation.

## 3. Materials and Methods

### 3.1. Apparatus

The experiment data were collected, including operating performance data; the variables of the fixed-base driving simulator used in this experiment were acceleration, deceleration, and lateral position. The values of gas and brake pedal use, representing the pressing intensities of these two pedals, ranged from 0 to 1. For instance, “0” denoted that the pedal was not pressed, while “1” denoted that the pedal was pressed to its full depth. The driving speed and gas pedal power were collected every 10 m before the sign, and that the values were averaged for all participants.

#### 3.1.1. Eye Tracker

In the experiment setup, a pair of Tobii Pro Glasses 3, consisting of a head-worn eye tracker connected to the recording device holding the saved data of the Tobii Pro Glasses 3, was used. This is a binocular eye-tracking setup that uses two cameras and six glints per eye for gaze tracking, and it was selected because it is known to be the most accurate eye-tracking unit [47,48,49].

The Tobii Pro Glasses recording device was connected to a desktop running the Tobii Glasses Controller software (version 1.11.4) (Tobii Technology, Stockholm, Sweden) using an ethernet cable. The system was set to the 50 Hz mode and calibrated with the one-point calibration procedure using the marker provided with the eye-tracking setup. The calibration consisted of (a) fixing the marker to the center of the stimulus grid, (b) instructing the participant to fixate on the marker’s center, and (c) entering the calibration mode in the Tobii software (Tobii Technology, Stockholm, Sweden), after which the process was completed automatically. The front-facing cameras recorded a video stream at 25 Hz with a resolution of 1920 × 1080 px, and the four eye cameras recorded a stream at 50 Hz containing four eye images (two views of each eye) with a combined resolution of 240 × 960 px. The study followed the Tobii Manual definition of fixation, which lasts more than 60 ms [50]. Fixation was defined as the time between the end of one saccade and the beginning of the next saccade.

#### 3.1.2. Driving Simulator

The UC-win/Road program was initially used for interactive virtual reality (VR) modeling in construction planning, urban planning, civil engineering, and traffic modeling [51,52,53]. By combining the program with a driving simulator, several driving situations in diverse environments can potentially be emulated. In this study, the experiment employed an automatic car comprising a seat, a seat belt, a steering wheel, a speedometer, an ignition and key unit, an accelerator, and a brake pedal. The visual display system consisted of three-color monitors spanning a 180° field of view synchronized to display a realistic view of a computer-generated road environment, as shown on Figure 1. Moreover, the simulator included an audio system to provide realistic traffic sounds and instructions.

The route comprised two main sections, i.e., curved and tangential. The driving experiment was performed in daylight under gloomy weather conditions.

### 3.2. Experimental Stimulus

Before the main simulator experiment, a preference survey of six static warning sign was conducted. This was an online survey in which 124 participants answered which of the six design concepts was the most preferred as a black ice warning sign. The more preferred design for the sign, which is shown in Figure 2a (top left), was selected [54].

After the preference survey of the sign design, the sign used in the experiment was redesigned using the same concept. The final design was modified to be actuated under freezing temperatures to show text messages with a pictogram for black ice. The pictogram was also revised to display the black ice warning even without the snowflake pictogram, as shown in Figure 2b. Detailed designs of all the signs mentioned in the study are shown in Figure 2.

### 3.3. Scenarios

The roadway geometry depicted in the simulations was a virtual-reality reconstruction of a real rural two-lane road, including horizontal curve and tangent segments, in Gangwon-do province. The roadway segment was a 5.56 km section of South Korea Route 28. This reconstruction was based on the topographic layout, which consisted of a straight road. The lane widths, road markings, sight distances, and other road engineering characteristics of the roadway segment, including a 3.6 m lane width and 140 m curve radius, were incorporated into the simulation to provide a similar road perception. Several different scenarios were designed with several combinations of experimental conditions and implemented in virtual reality. Two scenarios for geometric conditions, including a curved segment and tangent segment, and four scenarios for warning signs for black ice were used for the experiments. In each case, the traffic was displayed in random order. The signs were located 22 m after the end of curve segment and at the end of the 200 m tangent segment. Road markings were yellow and white, with a width of 10 cm. The length of road influenced by signs on the straight segment was 200 m. The corresponding length of road on the curved segment was 75 m.

In the four experiments involving signs, i.e., no sign, conventional static warning sign for black ice, new warning sign not actuated by weather conditions, and new warning sign actuated by weather conditions, as shown in Figure 3, two typical sign locations, i.e., at the ending of a curve section and at a point in a tangent section, were considered to investigate their effects on driving behavior, as illustrated in Figure 4.

### 3.4. Procedure of the Experiments and Participants

Each participant was tested individually in the experiment. As a first step, upon their arrival, the participants were briefed on the requirements of the experiment. All participants read and signed an informed consent document and were asked to complete a short questionnaire to provide basic information (i.e., age, gender, etc.). To collect concrete data for improving static warning signs for black ice, the participants were informed how the signs signify a freezing temperature with a red pictogram when actuated, as described in Section 3.2. As a second step, the participants were introduced to the driving simulator and provided an overview of its functions. Then, they were familiarized with the vehicle in a neutral condition. In this condition, each participant drove through a reconstruction of scenario 1 in reverse, which allowed for familiarization with the torque-based tactile feedback provided through the steering wheel and pedal control.

The participants were informed that they were to drive on a mountainous road until asked to stop. The participants were asked to adapt their speed to the driving conditions as follows: “Please drive as you would drive in the same situation in the real world until you are advised to stop. The speed limit on this roadway is 60 km/h”. The participants were informed that some disturbances could occur during the simulation and that they could stop the experiment at any time.

Next, the participants started to drive using the simulator for four separated runs in random order after completing calibration of the eye tracker. In each run, they encountered randomly assigned sign conditions. These experimental sign conditions included no signs, conventional static warning signs, new warning signs not actuated by weather conditions, and new warning signs actuated by weather conditions. All the participants drove in all situations while wearing the eye tracker.

After the fourth run, the participants were asked to complete a questionnaire concerning the effectiveness of each sign in terms of hazard perception in black ice zones. The participants took an average of 50 min to complete the experiments, with the driving phase lasting an average of 30 min

In this study, the experiment recruited 37 healthy subjects from our panel who occasionally joined our simulation experiment to participate in a driving simulator experiment with visual perception of traffic scenes. All subjects had a valid driver’s license and normal or corrected-to-normal vision (using contact lenses or glasses). Those who showed any signs of simulator sickness were excluded from the experiment. All subjects reported driving daily and holding a driver’s license for at least 3 years. Table 1 summarizes the gender and age variation of the subjects.

### 3.5. Data Analysis Methodology

#### 3.5.1. Determination of Location at Which Driver Behavior Starts to Change

To find the distance from the signs where drivers changed their behavior due to perception of the signs, vehicle speed and gas pedal power were used to identify significant differences between the group that drove with no signs and the group that drove with signs, including conventional static warning signs and new warning signs not actuated and actuated by weather conditions. To determine the influence of traffic signs, the study compared the datasets of the two groups. The study analyzed the behavior of drivers approaching the signs, during the approach, and in two geometrical sections. The indicators selected for these analyses included the vehicle speed and deceleration rate measured using the pressure applied to the gas pedal. The data were input into a connected computer for analysis.

#### 3.5.2. Correlation between Drivers’ Behavior Using Deceleration and Visual Data

The study compared the deceleration behaviors of drivers between the two groups, separated by visual data, fixation count, and duration data, i.e., the group that fixed eyes on signs and the group that did not. The fixation visit data were used to distinguish whether the subjects had looked at the signs. As addressed in Section 3.1.1, fixation was defined as points at which the eyes were stationary and focused on an object for longer than 60 ms. This study did not intend to find statistically significant differences during test between the two groups but intended to observe the distinction between groups regarding differences in speed deceleration as a function of the sign.

This study aimed to investigate the effectiveness of two different types of signs. The *t*-test was utilized to find significant differences between each sign and driving without any sign.

## 4. Results

The study first determined the location at which the drivers started to significantly change their speeds and gas pedal power. The results provided excellent indications of where the signs influenced the drivers. As shown in Figure 5, the locations where the drivers’ behaviors started to change signified the distance from the signs where the drivers began to exhibit significantly different driving behavior considering the presence of signs. Thereafter, speed differences at these locations among the participants were computed to determine the effectiveness of warning signs in indicating an upcoming black ice hazard.

### 4.1. Determination of Location at Which Driver Behavior Started to Change

The distance from the signs where the drivers’ behavior showed significant change could be identified on the basis of the agreement between the drivers’ behaviors and the presence of signs. As the data indicate, in each case, promising results were obtained in terms of the location at which driver behavior started to change in terms of vehicle speed and gas pedal power. In case of the curve segment, at 90 m before the signs, drivers drove at lower speeds (*M* = 45.88, *SD* = 11.26) than those who traversed the same roadway without any speed warning signs for black ice (*M* = 49.91, *SD* = 7.55), *t* (147) = 2.6, *p* = 0.012. Moreover, the gas pedal power of the drivers presented with a sign (*M* = 0.23, *SD* = 0.19) was lower than that of the drivers who were not presented with a sign (*M* = 0.3, *SD* = 0.17), *t* (147) = 2.4, *p* = 0.02, as summarized in Table 2. Meanwhile, in the tangent segment, at 160 m before the signs, the drivers who were presented with a sign drove at lower speeds (*M* = 55.84, *SD* = 7.93) than those who were not presented with any speed warning sign for black ice (*M* = 59.15, *SD* = 6.33), *t* (147) = 2.325, *p* = 0.02. Moreover, the drivers who were presented with a sign used less gas pedal power (*M* = 0.6, *SD* = 0.25) than those who were not presented with a sign (*M* = 0.72, *SD* = 0.17), *t* (147) = 3.282, *p* = 0.002.

Subsequently, speed and gas pedal power allowed us to distinguish the points to be analyzed to evaluate the effectiveness of warning signs in each case. On the basis of these comparisons, the location at which driver behaviors started to change for each sign were determined to be 90 m in case of the curve segment (*p* = 0.012) and 160 m in case of the tangent segment (*p* = 0.02).

### 4.2. Driving Behavior: Speed Deceleration

During determination of the location at which driver behaviors started to change in response to a sign, the maximum and minimum vehicle speeds were extracted to represent the effects of signs on drivers’ deceleration behaviors. Both indicators measured not only the intensity of drivers’ deceleration but also the overall behavioral shift in response to the signs. The deceleration value of each participant was computed by subtracting the minimum vehicle speed from the maximum vehicle speed. The means of drivers’ deceleration behaviors are summarized in Section 4.2.1 and Section 4.2.2.

#### 4.2.1. Conventional Static Warning Signs for Black Ice

Table 3 shows the results of multiple comparisons of average drivers’ deceleration behaviors in response to each sign. As a result, it was found that the conventional static warning signs did not lead to a significant difference in drivers’ deceleration behaviors in all cases (curve: *M* = 6.48, *SD* = 4.70; tangent: *M* = 7.54, *SD* = 5.50) relative to the no-sign case (curve: *M* = 7.83, *SD* = 4.94; tangent: *M* = 6.96, *SD* = 3.65). These results indicate that the conventional static warning signs did not significantly affect drivers’ deceleration behaviors, regardless of road geometry. Conclusively, the drivers were not significantly influenced by the conventional static warning signs for black ice.

According to Figure 6 and Figure 7, the values of vehicle speeds and gas pedal power were consistent between the cases where drivers were provided conventional static warning signs for black ice (old sign) and no sign at all. This result is in line with earlier predictions, i.e., static warning signs failed to provide a hazardous alert to drivers such that they could perceive the urgency to decelerate the vehicle. Therefore, there were no significant differences in drivers’ deceleration behaviors upon the provision of static warning signs for black ice.

#### 4.2.2. New Warning Signs Actuated by Weather Conditions

As can be inferred from Table 3, drivers exhibited enhanced deceleration behaviors when the new warning signs for black ice were actuated (not actuated/actuated curve; segment: *M* = 11.35/11.19, *SD* = 5.64/7.19; tangent segment: *M* = 14.72/11.11, *SD* = 7.95/7.64) than those when presented with conventional static warning signs for black ice (curve: *M* = 6.48, *SD* = 4.70; tangent: *M* = 7.54, *SD* = 5.5). This is because the preceding vehicles were affected only by the new warning signs actuated by weather and their deceleration behaviors changed.

According to Figure 4 and Figure 5, the speed and gas pedal power of the drivers presented with the new warning signs for black ice actuated by weather were lower, meaning that the new warning signs significantly influenced drivers’ deceleration behaviors. An interesting result was that the speed and gas pedal power values of the drivers presented with the new weather-actuated warning signs black ice were lower than those of the drivers presented with the new warning signs for black ice not actuated by weather. Thus, the warning information received by the drivers from the actuated signs had more significant effects on their vehicle deceleration behaviors.

By contrast, the results of all the new actuated warning signs for black ice in cases of the curve and tangent segment, summarized in Table 3, indicate that the vehicle had statistically significant differences in speed deceleration from the other signs, including the old sign and non-actuated sign, as well as no sign. The results in Figure 6 show that the average speeds associated with the new weather-actuated warning signs for black ice on the curve segment, in which the design of the new signs was similar to that shown in Figure 1b, were relatively lower than the average speeds associated with the new warning signs for black ice not actuated by weather on the curve segment. Moreover, in the case of the tangent segment, where the drivers were able to identify signs from 160 m, the gap between the vehicle speeds associated with the new warning signs for black ice actuated and not actuated by weather was smaller. This was because the preceding vehicle was affected by the warning information in the tangent segment, but the subject vehicle was affected by both the warning information and the change in maneuvering behavior in the curve segment.

Conclusively, the new warning signs provided the greatest safety benefit in terms of speed reduction. More detailed meaningful findings were identified by analyzing the results obtained with the two road geometries. First, the new warning signs for black ice were the most effective when they were actuated by weather, regardless of the road geometry. Second, geometrical differences had a positive effect on drivers’ speed reduction behaviors, regardless of whether they received warning information from different signs. That is, when the preceding vehicle was driving on the curved segment, the subject vehicle was affected by both the warning information and the change in maneuvering behavior in the curved segment.

### 4.3. Visual Behavior: Total Fixation Visit and Average Fixation Duration

The fixation visits, i.e., the number of driver eye fixations on warning signs, were measured as shown in Table 4. Fixation visits may represent the extent of an increased processing attempt for information [41]. The results show that the conventional static warning signs showed the lowest number of fixation visits in case of the curved segment (25 visits) and the tangent segment (65 visits). The new non-actuated warning signs showed 67 fixation visits in the curved segment and 93 fixation visits in the tangent segment. The new weather-actuated warning signs showed 62 fixation visits in the curved segment and 133 fixation visits in the tangent segment. These results indicate that the new weather-actuated warning signs had the most increased processing time, regardless of the road geometry.

Meanwhile, fixation duration, which indicates the processing time for which drivers’ gaze is fixed at one location, may also represent the increased processing demands for driver comprehension of each sign [41]. Fixation duration of the new warning signs was the longest in the curved segment (352.03 ms) and in the tangent segment (621.51 ms).

### 4.4. Correlations between Variables

Table 5 summarizes the drivers’ deceleration measure, standard deviation, and gas pedal power between two groups, i.e., the group that was presented with warning signs and the group that was not presented with any warning signs. The deceleration measure indicates the vehicle speed reduction between the starting point of change in drivers’ behavior and the point at which the signs were installed. The standard deviation shows the diameter of driver’s deceleration measure for each sign. The gas pedal power shows the pressure applied by the driver between the start of change in driver behavior and the point at which signs were installed.

Conventional static warning signs for black ice

The result of the speed deceleration comparison between the groups who were presented with the conventional static warning signs for black ice and no sign show the ineffectiveness of the conventional static warning signs in terms of reducing vehicle speed for safety, regardless of sign visibility. This means that the drivers who saw the conventional static warning signs for black ice drove the same as the drivers who did not see the signs. An interesting result was that the speed deceleration value in the tangent segment was higher than that in the curved segment, regardless of whether the drivers saw the signs.

2.New warning signs not actuated by weather

The results in Table 5 for the new non-actuated warning signs indicate that the new warning signs were more effective than the conventional static warning signs in terms of slowing down vehicles, regardless of whether they were actuated. An interesting result was that the speed deceleration value of new non-actuated warning signs in the tangent segment was higher (not seen: 13.30 km/h; seen: 15.99 km/h) than that of the new weather-actuated warning signs (not seen: 11.36 km/h; seen: 10.95 km/h). This is because the preceding vehicle was affected only by the new warning signs from the location at which drivers’ behavior started to change, and the distance between the sign and the location was 160 m, as shown in Table 2. The result indicated that drivers did not show a consistent pattern for the new non-actuated warning signs.

3.New weather-actuated warning signs

Previous studies on roadside signs have suggested that horizontal curves are among the most hazardous situations for drivers [17,18,19,20,21]. In most studies, drivers who are regularly unaware of approaching changes in roadway geometry or do not sufficiently reduce their operating speed when negotiating these geometric changes were analyzed [38,39,40]. Therefore, the new weather-actuated warning signs for black ice must demonstrably provide warning information to driver such that they can take appropriate actions when faced with a hazardous curve.

The results obtained using the new weather-actuated warning signs, summarized in Table 5, indicate that, in terms of slowing down vehicles or warning drivers about hazards in curved segment, these warning signs were the most effective when actuated. The speed deceleration due to the new weather-actuated warning signs increased by 33.7% when drivers looked at the signs (12.80 km/h) compared to when they did not look at the signs (9.57 km/h). Moreover, among all signs, the drivers who saw the new weather-actuated warning signs had the lowest gas pedal power (0.37/0.33) in the curved and tangent segments. This is because the preceding vehicle perceived the hazard warning from the new weather-actuated warning sign, which caused the driver of that vehicle to reduce the pressure on the gas pedal in response.

## 5. Discussion

Black ice is one of the main causes of winter traffic accidents, and warning signs for black ice are generally used to prevent those accidents. However, these signs provide warnings about black ice to drivers throughout the year, even in summer when the possibility of black ice is extremely low, because the sign is a static sign. Thus, drivers grow accustomed to the idea that the presence of the conventional static warning sign does necessarily indicate the presence of a hazardous condition ahead, causing drivers to ignore warning signs. To overcome this limitation, new warning signs for black ice were developed in a study under the Ministry of Land, Infrastructure, and Transport of Republic of Korea [55]. In that study, various technologies were used to change the color of the sign in response to different temperatures, and our research team investigated the performance and effectiveness of the new warning signs in terms of driver behaviors and deduced guidelines for installation of the developed signs.

A driving simulator was used to evaluate the effects of the conventional static warning signs and new weather-actuated signs for black ice for two different road geometries. The participants encountered a curve in a simulated road that offered a clear sight of the roadside. It was hypothesized that, upon provision of the new weather-actuated warning signs, the drivers would be cautious and decelerate to drive safely. Drivers’ behaviors were recorded after computing vehicle speed and gas pedal power data to determine where the signs had an impact, regardless of the geometrical aspect. The results indicated that the conventional static warning signs for black ice did not appropriately alert drivers. These results are consistent with the results of previous studies, which suggested that the static warning signs for occasional hazards may lose their effectiveness because no real threat is perceived by drivers most of the time [12]. This result is in line with those of previous studies in which static warning signs performed inadequately [12,13].

We systematically evaluated the effectiveness of warning signs for black ice by analyzing the behavioral changes of participants in different geometrical cases and the interactions between the participants and the installed signs in terms of speed deceleration values and gas pedal power. Furthermore, we investigated drivers’ behavior within visual correlation between the drivers who fixed and did not fix eyes on the signs. The results indicated that the existence of warning signs was significantly associated with drivers’ behaviors such as speed reduction. Moreover, this finding is consistent with the reviewed previous studies in which it was concluded that visual traffic warning devices are correlated with speed reduction [16,17,25,28,31].

This finding reinforces how drivers’ speed reduction suggests the effectiveness of warning signs. The conventional static warning signs, which had the lowest number of fixation visits and shortest duration, showed that drivers did not reduce their speed. Meanwhile, the new weather-actuated warning signs, which had the greatest number of fixation visits and longest duration, showed the most significant speed reduction. The increase in fixation visits and duration of new weather-actuated warning signs may represent that the signs warned drivers to become cautious, causing them to increase processing demands with respect to the signs [56,57]. Through the study, it was also found that when drivers were more fixed on the signs, they mostly presented greater speed reduction. This finding implies that traffic warning systems including warning signs must draw drivers’ eyes as a fixation maneuver. However, duration of fixation is related to ease of comprehension; thus, a higher duration of fixation may denote a lower efficiency of the signs and less safe conditions.

There are some limitations of this research to be improved through further studies. First, a study using a driving simulator can always have problems related to completely representing real driving situations, despite the many strengths of driving simulator experiments. The simulator was designed to recreate the characteristics of different driving conditions as precisely as possible, but it cannot capture the dynamism of real-life driving. Nevertheless, this simulator has been used extensively [31,32]. Second, Because South Korea does not permit a driver’s license for color-blind individuals, the experiment did not consider drivers with color blindness to analyze the effectiveness of black ice warning signs design. Third, to obtain more universal results, the sample size from the experiment may not be enough, and there was no consideration of population distribution including age and gender in Korea. Therefore, further research with larger samples, different ages, and similar proportions of individuals from different genders is needed to confirm the findings of this study [8,54,58,59] Fourth, as addressed in Section 3.1.1, fixation was defined as the point at which the eyes were stationary and focused on an object for longer than 60 ms [41]. Therefore, drivers who looked at warning signs for less than 60 ms were included in the group that did not fix eyes on signs. Further research with more appropriate regulation with correlated references for the definition of fixation are needed to confirm the findings of this study. Fifth, for the purpose of the study, participants learned the meaning of the new sign and mechanism of the actuated sign before beginning the driving simulator experiment. This may have brought some distortion of results. In further study, this issue should be considered carefully. Further investigations are also needed to determine the effect of such sign during nighttime, snow, fog, and rainy conditions.

## 6. Conclusions

Conventional static warning signs for black ice are commonly used in traffic environments worldwide [12]. Studies have demonstrated that many static warning signs have not been adequately evaluated to judge their effectiveness [12,13]. Even though the availability of warning signs is significantly associated with drivers’ deceleration behavior, these finding seem to suggest that the conventional static warning signs for black ice fail to clearly convey to a driver the hazardous conditions ahead with the lowest number of fixations and shortest duration among all signs. Lastly, the results of the experiments suggest that the static warning signs for black ice are statistically insignificant in terms of their effectiveness in providing hazard perception to drivers to reduce speed, with the highest number of fixations and longest duration time.

By contrast, the new weather-actuated warning signs successfully provided hazard perception to drivers, who reduced vehicle speed significantly in response. The new warning signs led to the greatest deceleration, which signifies their effectiveness in terms of clearly conveying hazardous conditions ahead to drivers. Moreover, the new warning signs were the most effective in terms of reducing vehicle speed when they were actuated by freezing weather. The number of fixation visits was the highest among all other signs. However, due to the unfamiliar new weather-actuated design, the sign showed the longest fixation duration [55,56]. Nevertheless, the new warning signs were associated with the lowest gas pedal power when they were actuated by weather.

This study not only provided valid analytic data about the effectiveness of black ice warning signs, which suggest that static conventional warning signs are inadequate while the new weather-actuated warning signs are adequate, but also exploited the benefits of the driving simulator and eye-tracking technologies, which could contribute to future evaluations of the effects of warning signs on drivers’ speed-reducing behaviors and visual attention toward signs.

## Figures and Tables

**Figure 1 ijerph-19-07549-f001:**
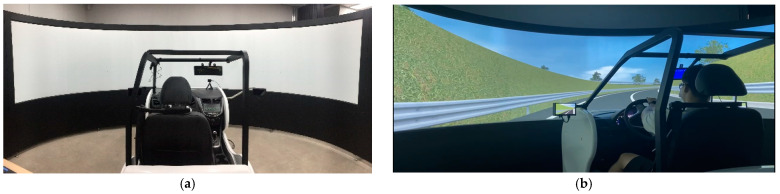
Location of study sites and the signs on the segments for experiments: (**a**) driving simulator; (**b**) driving simulator during experiment.

**Figure 2 ijerph-19-07549-f002:**
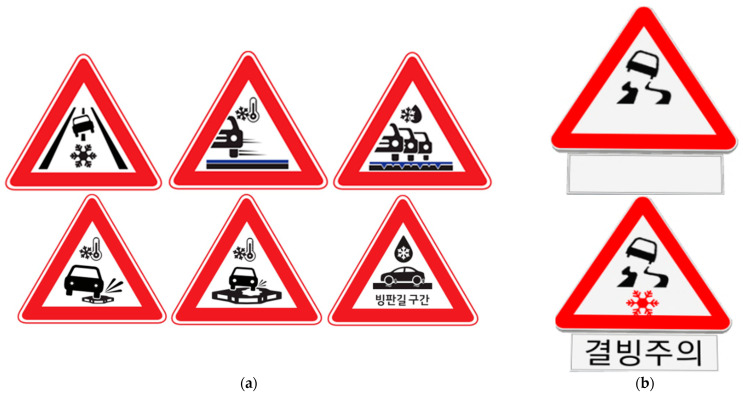
Study signs for each experiment step: (**a**) six versions of new warning signs actuated by weather conditions; (**b**) final selected design for new warning sign before actuation (**top**) and after actuation by freezing temperatures (**bottom**); The Korean sign means “Caution on Black Ice”.

**Figure 3 ijerph-19-07549-f003:**
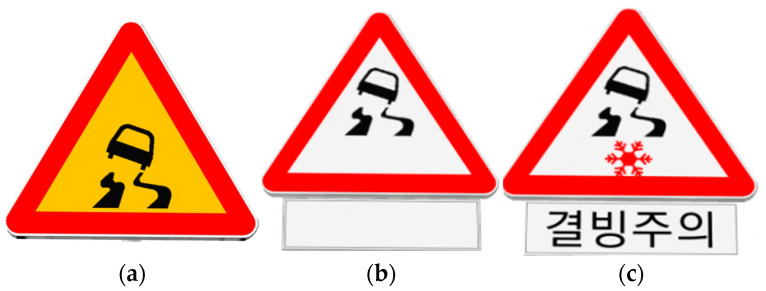
The signs used for VR simulator experiments: (**a**) conventional static warning sign for black ice; (**b**) new warning sign not actuated by weather conditions; (**c**) new warning sign actuated by weather conditions. The Korean sign means “Caution on Black Ice”.

**Figure 4 ijerph-19-07549-f004:**
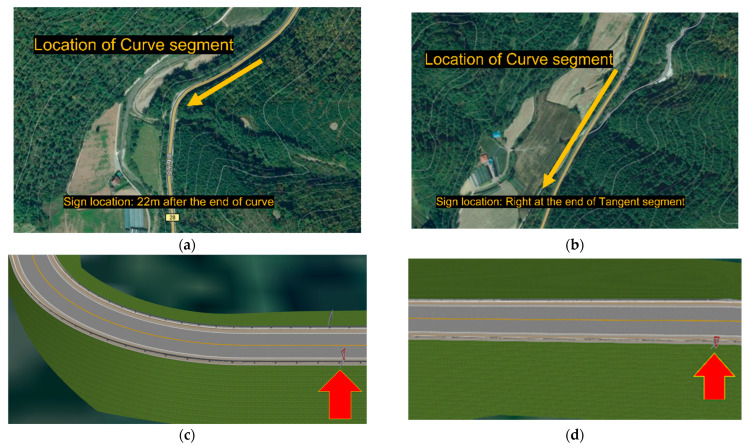
Location of study sites and the signs on the segments for experiments: (**a**) curve segment study site; (**b**) tangent segment study site; (**c**) curve segment in experiments; (**d**) tangent segment in experiments.

**Figure 5 ijerph-19-07549-f005:**
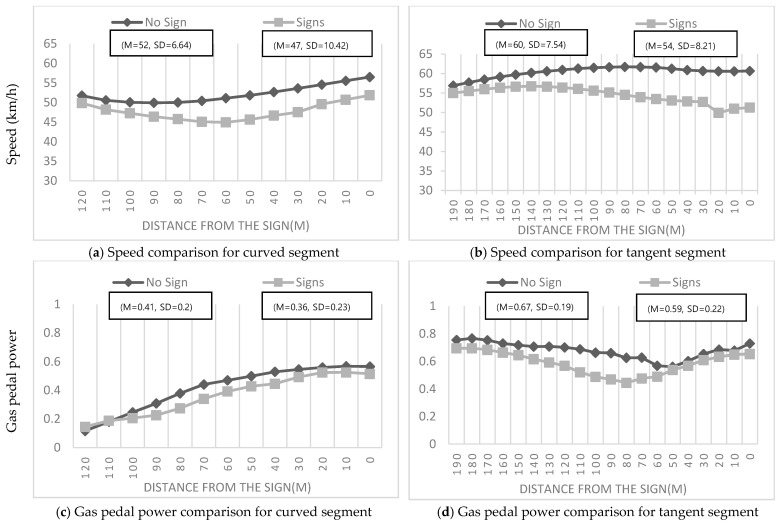
Speed and gas pedal power comparison for no-sign conditions and with signs in curve segment (**left**) and tangent segment (**right**).

**Figure 6 ijerph-19-07549-f006:**
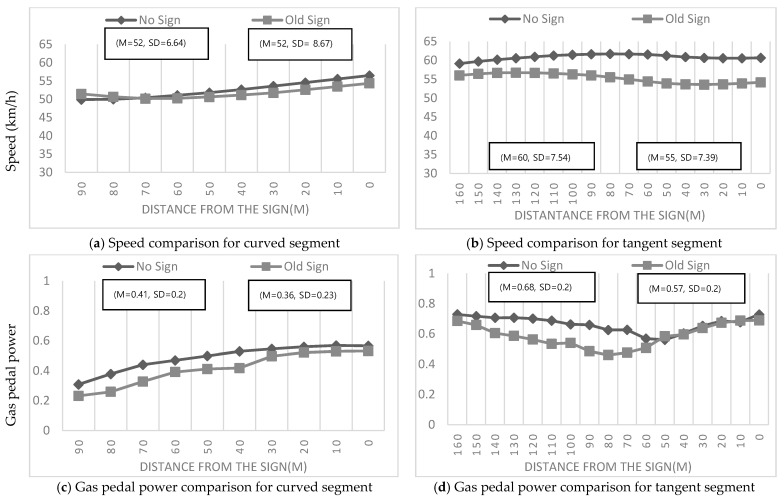
Comparison between no-sign conditions and conventional static warning signs for black ice.

**Figure 7 ijerph-19-07549-f007:**
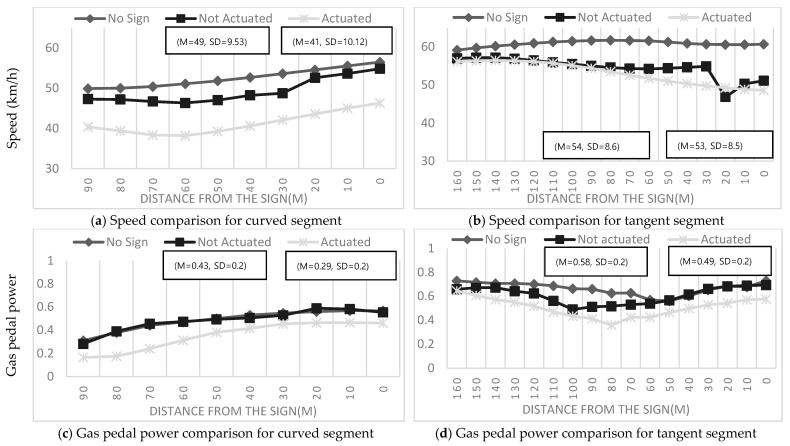
Speed and gas pedal power comparison between no-sign conditions and each sign, including conventional static warning sign and new warning signs for black ice actuated and not actuated by weather.

**Table 1 ijerph-19-07549-t001:** Participant age groups by gender.

Age	20s	30s	40s	50s	Total
Male	4	13	6	6	29
Female	2	2	3	1	8
Total	6	15	9	7	37

**Table 2 ijerph-19-07549-t002:** Speed and gas pedal power comparison between no-sign conditions and signs in *t*-test for two cases.

Distance From Sign (m)		Mean (Standard Deviation) No Sign/Signs		*t*-Value (*p*-Value)	
		Speed	Gas Pedal Power	Speed	Gas Pedal Power
Curve segment	120	51.74/49.17 (8.74/11.37)	0.12/0.15 (0.15/0.18)	1.482(0.143)	−1.083 (0.283)
	110	50.55/47.68 (8.55/11.51)	0.19/0.19 (0.16/0.19)	1.4 (0.163)	0.053 (0.958)
	100	50.05/46.74 (8.10/11.42)	0.25/0.19 (0.17/0.19)	2.001 * (0.049)	1.653 (0.104)
	90	49.91/45.88 (7.55/11.26)	0.3/0.23 (0.17/0.19)	2.565 * (0.012)	2.395 * (0.02)
Tangent segment	190	56.89/56.96 (6.06/7.31)	0.75/0.67 (0.13/0.18)	1.463 (0.145)	2.627 ** (0.009)
	180	57.70/55.37 (6.13/7.48)	0.76/0.66 (0.12/0.2)	1.725 (0.086)	2.888 ** (0.004)
	170	58.47/55.69 (6.23/7.64)	0.75/0.63 (0.12/0.23)	2.019 * (0.045)	4.257 ** (0.000)
	160	59.15/55.84 (6.33/7.93)	0.72/0.6 (0.17/0.25)	2.325 * (0.021)	3.282 ** (0.002)

* *p* < 0.05, ** *p* < 0.01.

**Table 3 ijerph-19-07549-t003:** Comparison between each of the signs in terms of speed deceleration values in two cases.

Sign Condition		*t*-Value (*p*-Value)			
		No Sign	Old Sign	Not Actuated	Actuated
Curve segment	No sign	-	-	-	-
	Old sign	−0.048 (0.962)	-	-	-
	Not actuated	−0.426 (0.672)	−0.360 (0.72)	-	-
	Actuated	−3.317 (0.001 **)	−3.125 (0.003 **)	−2.683 (0.009 **)	-
Tangent segment	No sign	-	-	-	-
	Old sign	−1.220 (0.227)	-	-	-
	Not actuated	−0.192 (0.848)	1.103 (0.274)	-	-
	Actuated	−2.978 (0.004 **)	−1.878 (0.065)	−2.934 (0.005 **)	-

** *p* < 0.01.

**Table 4 ijerph-19-07549-t004:** Fixation visits and fixation duration for each sign in two cases.

		Number of Participants Who Fixed Eyes on Sign (%)	Total Number of Fixation Visits (N)	Mean of Fixation Duration (SD) (ms)
Curve segment	Old sign	12 (32%)	25	252.36 (222.9)
	Not actuated	19 (51%)	67	247.27 (162.5)
	Actuated	19 (51%)	62	352.03 (239.0)
Tangent segment	Old sign	16 (43%)	65	499.44 (594.9)
	Not actuated	19 (51%)	93	621.51 (782.7)
	Actuated	23 (62%)	133	606.67 (715.6)

**Table 5 ijerph-19-07549-t005:** Comparison between groups who fixated or did not fixate on signs.

		Eye Fixation on Signs/No Eye Fixation		
		Old Sign	Not Actuated	Actuated
Curve Segment	Speed deceleration (km/h)	6.84/6.32	10.83/11.92	12.80/9.57
	Speed standard deviation (km/h)	2.4/2.21	4.08/4.44	4.49/3.34
	Gas pedal power	0.38/0.43	0.50/0.48	0.37/0.33
Tangent Segment	Speed deceleration (km/h)	8.15/7.04	15.99/13.30	10.95/11.36
	Speed standard deviation (km/h)	2.84/2.60	4.55/4.31	3.93/3.97
	Gas pedal power	0.62/0.56	0.64/0.58	0.53/0.5

## Data Availability

The data used to support the findings of this study are available from the corresponding author upon request.

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
