# Peer review of "Improvements of Warning Signs for Black Ice Based on Driving Simulator Experiments"

_ijerph, 2022, doi:10.3390/ijerph19127549_

Round 1
Reviewer 1 Report
In the abstract, I would suggest trying to present actual results and their direction instead of a general comments "were clearly observed in the experiments".
line 65, do not (no not)
line 79 and 80: I don't know if this sentence is a continuation of the above paragraph but it seems ut of place or lacks other sentences and references to be fully used.
line 111 repeats inexperienced and makes the sentence incomprehensible
line 115, why not considering the brake pedal as well
line 119, should increase
line 143: and pedals' control
Were there any signs of simulator sickness reported by participants? This should be noted since it might influence driving behaviors (Mackrous et al. 2014).
There is also more information to be added regarding the simulator such as the model, the field of view, etc. to allow for a better understanding of the experimental set-up and for future comparisons of the results obtained here. the sentence on line 176 should describe directly the simulator instead of referring to other papers for readability.
I suggest that figure 2 depicts the same road (bird's eye view with the different locations) since it is hard to evaluate the difference between both presentations with a different map overlay.
line 207 is missing a dot(.)
line 171: Figure 1 (instead of Figures 1a,b)
line 228 repeats the information provided above and could be removed
revise the sentence at line 248 because it is unclear what is the data processing associated to it
line 252: visual search data (instead of visual data)
line 254: are there subjects who have not look at the sign at all ? This section should also specify how participants split into these 2 groups.
Figure 3 should resent data variability (SD or SE) and the title has to be revised.
section 4.1: authors present statistics only for one distance, what is the decision supporting this? How about closer distances?
Statistical analysis should also be presented at the end of the method instead of being presented directly in the results section. Moreover, a justification of the t-test should be done since the Table 2 suggest a multiplication of such a test and that an ANOVA or another choice could have been made. Table 3 would automatically fall into this consideration using a posthoc test.
I would suggest adding more details of the results obtained from Table 3 in section 4.2.
The start of 4.2.1 is actually a repetition of 4.2 and should be merged.
The purpose of color and shape for traffic signs is to clearly convey information that is coherent with one another. However, and this would have to be clarified in the current manuscript, how one who is color blind would be able to appreciate such a change in color? Considering it is a common disability, this can't be left aside as an implementation issue.
Overall, the results get confusing with the multiplication of figures that are presented differently. Some of the x axis are inconsistent between figures and makes it hard to compare conditions and results.
Table 4 have to present variations of the mean.
Though the topic is of great interest, I believe there should be improvements to the results section to better present and compare the various conditions and characterizations of the groups based on the visit or not of the signs.
Author Response
"Please see the attachment."

Reviewer 2 Report
This submission deals with an important topic: how to best provide warnings for intermittent roadway hazards (e.g., black ice, in this instance). It is generally very well written, though there are a few instances where improvements can be made. All comments are listed below:
- Line 110 – “One study calculated braking responses, number of fixations (N), and fixation duration (milliseconds) among experienced and inexperienced drivers. The results indicated that hazard perception was faster among inexperienced drivers than among inexperienced drivers [44].”
- Two “inexperienced” here
- Line 150 – “Next, the participants started to drive using the simulator for four separated runs in random order after completing calibration of the eye tracker. In each run, they encountered randomly assigned sign conditions. These experimental sign conditions included no sign, conventional static warning signs, new warning signs not actuated by weather conditions, and new warning signs actuated by weather conditions.”
-
- Fine as is, but this sort of experimental design information can perhaps be better displayed/understood in tabular format – consider this for any subsequent iterations of the submission
-
- Line 255 – “This study did not intend to find statistic differences of speed deceleration from the other signs including old sign and sign not actuated between the two groups but, intended to observe the distinction between the groups regarding differences”
- Missing punctuation and poorly placed punctuation
- The authors state what they are not looking for (or what they do not expect to see), but they don’t state anything about the nature of the differences they are looking for or do expect to see, or anything about the analytical techniques that will be deployed in such an effort – a bit more explanation and clarification here is warranted
- Line 448 – “The results obtained using the new weather-actuated warning signs, summarized in Table 5, indicate that in terms of slowing down vehicles or earning drivers about hazards in curved segment,”
- Earning appears wrong – please address
- Line 490 – “This finding reinforces why driver’s deceleration value was higher in the tangent segment, thus suggesting that greater numbers of fixation visits and longer fixation durations are positively related to vehicle speed reduction.”
- This sounds like a positive result in terms of speed reduction; however, are increased number of fixations and longer duration fixations sufficiently high to result in concerns related to driver distraction? This idea should be addressed in some fashion in the Discussion
- It is good that the authors discuss the strengths and limitations of the driving simulator itself; however, the ability of the driving simulation to recreate actual weather conditions is another topic altogether – in real life drivers would have some level of understanding of the weather – how cold is it? Is black ice possible, likely? Etc. Such factors may not have been captured in this study, but they would be present in real life.
- The entire Conclusions section is redundant, but I suppose it could be argued that’s its purpose. However, the paragraph in the Conclusions section beginning on line 534 appears to be entirely redundant with other material in the Conclusions section – if so, it can be omitted
Author Response
"Please see the attachment."

Reviewer 3 Report
The study investigated the impact of the performance and effects of the new warning sign for black ice on roads. The study was conducted on a driving simulator and included 37 participants. Overall, the study is interesting, however major improvements are needed before publishing. More detailed comments are presented below:
- Introduction, row 37: „Moreover, the coefficient of friction under such conditions is lower than those under dry and wet surface conditions, which increases the braking distance and the probability of traffic accidents.“ – I would suggest that authors add that lower coefficient of friction, except the increase in the braking distance, also affects the stability of the vehicle.
- Typoss in rows 63 and 65.
- „Travel speed and gas pedal power for acceleration before signs have been investigated to determine the effects of conventional static warning signs on drivers’ behaviors.“ – I do not see the need for this sentence, at least not where it is now. I suggest deleting it.
- Although it is clear what does it mean, usually it is prefered to use „traffic control devices“ instead of „traffic safety devices“ since road signs and road markings provide wider range of information to the drivers, not only those which are related to safety.
- There are many more studies which could fall under the section 2.2. However, due to the aim of the study, and the fact that it focuses on road signs, I would delete this section and instead expand the literature review on the effect of road signs on safety in different conditions – general effect. After that, I suggest that authors present what has been done in terms of the impact of signs in winter weather conditions. Available scientific literature in that sense is limited, however there are several reports published in the US which on the topic of the impact of warning signs for ice bridges etc.:
- Veneziano, Z. Ye, and I. Turnbull, “Speed impacts of an icy curve warning system,” IET Intelligent Transport Systems, vol. 8, no. 2, Mar. 2014, doi: 10.1049/iet-its.2012.0110.
- Lindgren and S. st. Clair, “Butte creek ice warning system,” Salem, Oregon, USA, 2009.
- Liu, N. Wang, H. Yu, J. Basara, Y. E. Hong, and S. Bukkapatnam, “Black Ice Detection and Road Closure Control System for Oklahoma,” Oklahoma City, OK, USA, 2014.
- Michigan Department of Transportation, “Bridge deck warning system, roundabouts highlight a year of MDOT safety projects ,” 2019.
- „The results indicated that hazard perception was faster among inexperienced drivers than among inexperienced drivers [44].“ – Please check the following sentence since it is not clear which drivers have faster hazard perception.
- „If drivers consider a sign to be necessary, the number of fixations and fixation duration time increase.“ – I would not agree 100% with this. If fixation time increases, this could mean that driver's did not understand the meaning of the sign and thus are fixating more and longer to figure this out - https://www.sciencedirect.com/science/article/pii/S0003687020301447
- Why is such disbalance between male and female participants? Moreover, instead of categorizing the age of participants („In terms of age, 6 participants were in their 125 20's, 15 participants in their 30's, 9 participants in their 40's, and 7 participants in their 50's“), authors could provide average age, min, max and std of age. In this way it would be easier for the reader to understand the sample size. In addition, could authors do the same for driving experience? It would be good to know what is the driving experience of the sample. Also, authors should describe recruiting methodology of participants.
- I would suggest moving the section 3.2.2. as a 3.1., section 3.2.1. as a 3.2., 3.2.3 as a 3.3., 3.2. as a 3.4. (without subsections) and then 3.1. as a 3.5. This seems more logical and easy to follow to me. Namely, readers would be introduced to the apparatus, sign and scenario designs and then experimental procedure. In current form, it is hard to follow the procedure since the reader is not introducted to the apparatus and sign and scenario design before.
- Figure 1 shows six different versions of the warning sign which were evaluated (a), and then on the b) parto f the figure it shows completely different sign design which has been used in the study. Why completely different sign design if there was an evaluation of several designs? I'm not saying that design b) is „bad“, but it is just unclear to me, why they differ from a) signs. Authors should clearly describe why specific sign is choosen.
- If I'm not mistaking, participants knew the aim of the study and the meaning of the signs was described to them before experiment. I'm wondering how do authors think this impacted participants behaviour and wouldn't it be better if they did not know that so their reaction could result from their instinct and not from the fact that they new exactly what are the conditions? Again, I'm not saying this is „wrong“ but it may have impacted the results and as such I feel it should be addressed at least in the discussion part.
- „The technical parameters of the fixed-base driving simulator used in this experiment can be found in [20,28-32]“ – I would appreciate if authors could write basic technical parameters of the used simulator and not refer readers to additional literature.
- „The lane widths, road markings, sight distances, and other road engineering characteristics were incorporated into the simulation to provide similar road perception.“ – Authors need to describe each of the following settings in order to enable reproductivity of the study.
- Authors should clearly describe the location of the signs, it's dimensions as well as the road geometry (cruve radii etc.).
- Authors should combine the text from rows 176-182 with the section 3.3.2. since it describes the driving simulator.
- Section 3.4.1. and 3.4.2. have several same or similar sentences. Overall, I do not see the need for subsection in section 3.4. In my view, authors could just elaborate what they measured and how. Also, in the data analysis section, authors should state which statistical tests were used.
- Row 270: „Analyses of the speed and gas pedal power of vehicles can be performed to determine whether the presence of signs influenced vehicle speed and gas pedal power.“ – Again authors are repeting the sentence which has already been stated.
- Data recording is to some extent confusing. Namely, authors state „In case of the curve segment, at 90 m before the signs, drivers drove at lower speeds (M = 276 45.88, SD = 11.26) than those who traversed the same roadway without any speed warning 277 signs for black ice (M = 49.91, SD = 7.55), t(147) = 2.6, p = .012).“ – the highest speed difference is at 60 m if I'm not mistaking when looking at the Figure 3. Do authors mean that 90 m is the first distance where there is a significant difference? Authors should in the previous section (3.4.) clearly explain that driving speed and gap pedal power were collected every 10 m before the sign and that the values were avereged (I assume they are) for all participants. As it is now, it is not clear how data was collected.
- Rows 326-329: it is not neccesary to write hypothesis again since the purpose of the study has already been described earlier.
- Figure 4 and 5– why all four cases are not represented on one grapf for each road geometry?
- Could authors write in text what were the average reductions in measured variables for each case so the reader could have the feeling of the manginute of the effect?
- Rows 379-382: again, authors do not need to repeat that since it was already stated what is the aim of the study.
- „These results indicate that the new weather-actuated warning signs were the most effective in terms of attracting attention, regardless of the road geometry.“ – Acctualy, the results indicate that non-actuated sign resulted in more fixations than actuated one in curve segment.
- „The fixation duration of the new non-actuated warning signs was the shortest (247 ms) in the curved segment and 478 ms in the tangent segment. These results indicate that the new non-actuated warning signs allow drivers to recognize that the hazardous condition is not likely to appear, which encourages drivers to divert their visual resources elsewhere.“ – I dissagree with what has been written. In curved section, participants had less time to look at the sign, due to the geometry, and thus had less fixations which lasted shorter compared to the tangent section. Authors should not focus on the comparison of curve and tangent since those are two different conditions in terms of geometry. Authors should compared sign conditions between each other in each road geometry. From the table 4, it can be seen that actuated sign resulted in longer average fixation duration compared to other cases. Authors should elaborate on that (in the Discussion section) and provide some explanation for this, is it due to the fact that participants did not know what does the sign mean, or due to the fact they were more careful and were checking the sign more and longer?
- I would suggest that authors combine section 4.4. with Discussion section (Discussion section should be it's own section and not a subsection of the results) and compare obtained results with other literature findings on the same/similar topic. Also, in the Discussion part, authors should elaborate on the reasons why such actuated signs are beneficial in terms of how humans perceive flashes etc. In other words, better description of WHY such results are obtained and WHY aforementioned signs affect drivers in such way is needed.
- Authors should add that the study was conducted in daytime conditions and that further investigations are needed to see the effect of such sign during night-time, when they are even more needed.
- I'm not a native English speaker, but there are some typos and it would be good to check and proofread the manuscript.
Author Response
"Please see the attachment."

Round 2
Reviewer 1 Report
the v2 received does not have Figure 3b and makes it impossible to identify if it is appropriate or not
Line 245 to 251 are hard to understand as to their appropriateness in the text. They look more like a historical perspective and a sale pitch, I suggest removing them.
The resolution of the figures have to be improved for readability.
In the response to reviewer: "Figure 3 should resent data variability (SD or SE) and the title has to be revised." < Response: As the reviewers’ comment, we recognize the lack of SD or SE, However, due to unclear analytic, we have decided to not include SD or SE on this paper.> This is not an appropriate response and such an issue should also be presented in the limitations. The authors present Mean and SD at line 308 so there is no reason such values are not depicted in the graphics.
Table 4 shows with the new information that about half of the participants did not look at the sign. Do they behave differently than other drivers? If 50% do not look at the sign, how can we be sure the sign is effective at improving road safety? This should be clearly explained and detailed in the discussion.
Overall, the paper has been improved but there is still a place for improvement to make the paper worthy of publication.
Author Response
the v2 received does not have Figure 3b and makes it impossible to identify if it is appropriate or not
<Response: After checking the PDF file, we realized that many of our Figures went missing during the process. We fixed the problem and make double-check.>
Line 245 to 251 are hard to understand as to their appropriateness in the text. They look more like a historical perspective and a sale pitch, I suggest removing them.
<Response: As the review comment, we agree to remove unnecessary information and relate it to an experimental component.>
The resolution of the figures have to be improved for readability.
<Response: The quality of review report-related Figures has been revised with higher resolution.>
In the response to reviewer: "Figure 3 should resent data variability (SD or SE) and the title has to be revised." < Response: As the reviewers’ comment, we recognize the lack of SD or SE, However, due to unclear analytic, we have decided to not include SD or SE on this paper.> This is not an appropriate response, and such an issue should also be presented in the limitations. The authors present Mean and SD at line 308 so there is no reason such values are not depicted in the graphics.
<Response: As the review comment, we added mean and SD in the Figures.>
Table 4 shows with the new information that about half of the participants did not look at the sign. Do they behave differently than other drivers? If 50% do not look at the sign, how can we be sure the sign is effective at improving road safety? This should be clearly explained and detailed in the discussion.
<Response: We agree with the reviewer's comment. Fixation is not an exact measurement to judge whether drivers look at the sign or not. Fixation was defined as points at which the eyes are stationary and focused on an object for longer than 60 milliseconds based on Tobii glasses' default condition. Therefore, drivers who looked at warning signs for less than 60 milliseconds were included in the group that did not fix their eyes on signs. Due to the VR driving simulator experiment, we compared the results based on various conditions. We investigated which sign systems are more effective. The following issue had been revised to be explained and detailed in the discussion>
Overall, the paper has been improved but there is still a place for improvement to make the paper worthy of publication.
Reviewer 3 Report
My comments related to the second stage of the review are in the pdf document.

Author Response
1. Sometimes authors use „road signs“, sometimes „Road-side sign“, while sometimes „traffic sign“. Please stick to one terminology for the sake of clarity.
<Response: The word had been revised to follow an official term on 2012 Supplement to the 2004 Edition of Standard Highway Signs of MUTCD as “warning signs”
2. I do not see significant difference between subsection 2.1. and 2.2. Both of them speak about impact of road signs on safety so why not combine both subsections into one? In general I think both 2.1. and 2.2. even if left separated should be improved with more „flow“ and context. For example, the title of 2.1. is „Static conventional warning signs...“, while in the row 75 authors write „Most studies that have investigated the effectiveness of warning signs have considered non-conventional warning devices...“ – why mentioned non-conventional warning devices if the title is related to conventional ones? Please rethink and try to improve aforementioned subsections.
<Response: The paper had been revised to combine both subsections into one as reviewer comment. The flow and context had also been revised to improve subsections.>
3. Rows 129-136: This is not literature review. Instead authors are speaking about the methodology of their study and as such, aforementioned paragraph should be placed somewhere under section 3. Also, since authors wanted to talk about hazard perception and visual behavior they should address more studies as a part of literature review.
<Response: The corresponding part had been revised to the methodology and the previous literature review was modified to focus more on visual behavior.
4. As stated in my comment in Review 1, authors should describe used simulatore (what are the components, number of screens, resolution, refreshing rate etc.). „acceleration, deceleration, and lateral position“ are variables that can be collected not technical characteristics of the simulator. This is important in order to understand the quality of the simulator and thus objectiveness of the obtained results. After further read, I see that more details are presented in subsection 3.5.2. I suggest that authors combine text from 3.5.2 and 3.1. for the driving simulator and also add another subsection under the „Apparatus“ for eye tracker (move 3.5.1). In general, driving simulator and eye tracker are Apparatus and not experimental stimuli (experimental stimuli are the used road signs).
<Response: Just as review suggested, the paper had been revised to combine text from 3.5.2 and 3.1. for the driving simulator and add another subsection under the “Apparatus” for eye tracker. We have stated acceleration, deceleration, and lateral position as variables for the revision.>
5. As stated in Review 1, „The lane widths, road markings, sight distances, and other road engineering characteristics of the roadway segment including 3.6m lane width, 140m curve radius were incorporated into the simulation to provide similar road perception.“ – what is the width and color of road markings, what is the distance of straight sections, what if the lenght of the curve etc. Authors need to in detailed describe the scenario in order to enable proper reproductivity of the study.
<Response: The following information has been revised to describe the scenario in detailed.>
6. Figure 1a is hard to understand to due to small dimensions of the picture. On the other hand, I cannot see Figure 1b, I don't know what is the reason for that.
<Response: The quality of related Figures has been revised with higher resolution.>
7. Regarding the participants, I do not understand why authors used ranges for age since there are no privacy issues regarding stating your date of birth if all other data privacy protocols are followed. I understand that impact of age and gender is not the focus of the paper, however, in the driving simulator studies it is recommended to have sample with characteristics relatively close to the general driving population of the country in order to have more objective results. Having said that, authors should definitely state the participants sample characteristics as a limitation of the study.
<Response: As the reviewer comment, the paper has been revised to state the participants sample characteristics as a limitation of the study>
8. Based on my comment in the Review 1, authors replied: „From our previous research, we found out that participant would not respond to signs if experiment do not specify the mechanism behind new warning sign and how it look like before the experiment. For the study, we decided to reconduct the experiment as giving a presentation of how new warning sign function and what it means before the experiment.“ – But if you tell participants what will happen then you are „priming“ them to some extent, and thus participants will focus mainly on this main task and thus be more efficient in executing it. If you don't tell them what is the aim, their attention will be „spread“ on different road/environment characteristics, and it will more closely resemd the real drive (mostly no one drives intentionally too „check“ some risky situation). If in such approach, participants do not respond to the signs, then maybe it is the problem of the signs and it's design. I feel that this is really important limitation of the study methodology and as such needs to be stated.
<Response: We agree with the reviewer comment. However, we explained just the new signs’ meaning and functions as primary education and advertisement of new transportation systems including traffic sign targeted at the public when they are implemented in the first times for increasing the effect of them and decrease erroneous action. We did not explain to participants what will happened and the specific experiment purpose before the experiment started as the reviewer point out. And experiments using VR simulators are not generally conducted to get results that we expect to happen on real roads, but they are conducted to get relative results with treatments or experimental conditions. Considering this prior notion and results of our previous experiments. Anyhow, we have revised to state the following limitation in discussion part.>
9. Subsection 5.1. should be under Results section and not under Discussion.
<Response: Subsection 5.1 had been revised to be under results section.>
10. Discussion must be improved as stated in the Review 1. I appreciate that authors improved it, but it is not sufficient and does not actually provide the discussion of the results. Example for this are following sentences: „. This finding reinforces why how driver’s deceleration value was higher suggest that greater numbers of fixation visits are positively related to vehicle speed reduction. Meanwhile, longer fixation duration might be highly related to difficulty of signs comprehension [53].“ – I'm really having trouble figuring out what authors wanted to say. Moreover, in the second sentence authors provide contrary explanation for the results and do not provide and elaboration on to what extent such „situation“is present in their study (at least their opinion). Overall, there is hardly any discussion on the results and connection of the results with literature.
<Response: We agree with the reviewer comment and there was some unclear explanation and logical conflicts. Therefore, we revised the paragraph to make sure and the avoid misunderstanding regarding the study result.>
11. In my opinion, a professional should check and proofread the manuscript.
<Response: We agree with the reviewer comment. However, due to the urgent deadline, we were unable to proceed proofread by professional. We will upload the final version of the manuscript after being proofread by an expert.>